# The Influence of ASD Severity on Parental Overload: The Moderating Role of Parental Well-Being and the ASD Pragmatic Level

**DOI:** 10.3390/children9060769

**Published:** 2022-05-24

**Authors:** Eva M. Lira Rodríguez, Rocío Cremallet Pascual, Miguel Puyuelo Sanclemente, Pilar Martín-Hernández, Marta Gil-Lacruz, Ana I. Gil-Lacruz

**Affiliations:** 1Faculty of Human Sciences and Education, University of Zaragoza, 22003 Huesca, Spain; rociocremallet@gmail.com (R.C.P.); puyuelo@unizar.es (M.P.S.); 2Faculty of Social and Human Sciences, University of Zaragoza, 40003 Teruel, Spain; pimartin@unizar.es; 3Faculty of Health Sciences, University of Zaragoza, 50009 Zaragoza, Spain; mglacruz@unizar.es; 4School of Engineering and Architecture, University of Zaragoza, 50018 Zaragoza, Spain; anagil@unizar.es

**Keywords:** autism, pragmatic language, parents, overload, well-being

## Abstract

The aim of the present study is to analyze the relation between the severity of symptoms in people with ASD on their parents’ overload, moderated by parental well-being and the ASD pragmatic level. A sample consisted of 28 fathers and mothers whose children had ASD. The obtained results showed that the higher the ASD severity, the better the parental overload was perceived if parents had low well-being levels. However, this relation did not occur if the parental well-being level was high. Moreover, the relation between severity and parental overload moderated by parental well-being occurred regardless of the pragmatic language level. Therefore, the main results of this study are that the responsibility for parental overload depends more on parental well-being than on the symptom severity of the person with ASD. The relevance of carrying out interventions with not only people with ASD, but also with their parents or caregivers for their well-being is highlighted.

## 1. Introduction

Autism spectrum disorder (ASD) has achieved substantial social visibility thanks to the efforts of many professionals, associations, families and institutions. Above all, this is thanks to the media’s influence in that media visibility acquires social relevance, allowing us to determine the scope of ASD’s diversity, repercussions and implications [1]. ASD is one of the most complex disorders to treat given the wide variability of its symptoms [2].

According to the Diagnostic and Statistical Manual of Mental Disorders (DSM-5) of the American Psychological Association [3], ASD is characterized as:
It is a neurodevelopmental disorder of neurobiological origin that begins in early development period stages […], characterized by restrictive and repetitive patterns of behavior, and by persistent deficiencies in social communication and social interaction. […] Symptoms cause clinically significant impairment in social, occupational or other important areas of normal functioning. […] Furthermore, these alterations are not better explained by other intellectual disability or by the global developmental delay”. (p. 50)

Ref. [4] considers the clinical picture that begins in early childhood and diminishes social and communication skills. Ref. [5] argue that ASD is characterized by patterns of behavior, activities or restricted interests manifesting more or less intensely depending on the subject’s capabilities and the intervention received. However, these authors also indicate that there are certain symptoms that are common in all cases. Regarding its prevalence, the data of the World Health Organization in 2021 estimate that 1 in 160 children has ASD. In recent years, various studies reveal a gradual increase in cases. This is, as [6] assure, possibly due to the changes in the diagnostic criteria of DSM-5 and the increased specialization by the professionals involved in the detection and intervention of the subjects.

The linguistic competence level of ASD is highly heterogeneous [7]. Most authors affirm that the acquisition and use of language are affected [7,8]. Ref. [9] indicates that in 50% of cases, subjects never develop functional language. The most palpable alterations are in the dimensions of conversational pragmatics and social communication [3]. Indeed, pragmatic difficulty has been associated with loneliness in the mothers of children with ASD or FXS, as well as depression, less satisfaction with life, and poorer family relationship quality in the mothers of children with FXS [10].

Many of the children with ASD require different intervention types (behavioral, language, social, occupational, pharmacological, or others), for which they attend specialized public or private care centers to receive therapies. During the COVID-19 confinement period, children were unable to continue their therapies. In some cases, this triggered alterations, and generated difficulties and overload in parents and/or caregivers [11]. In the study by Huang et al. (2021), parents reported that their children had: sleep problems (50.3%); improved cognitive ability, language expression and comprehension (40%); worsened emotional and social performance (36.2%); and decreased training intensity (60.8%). The most common dysfunctional behaviors observed in children with ASD were being easily distracted, losing their temper and crying. A total of 81.3% of parents had no anxiety, but 98% of parents reported that their family was under pressure [12].

The present study aims to assess the relevant aspects related to how the relatives of children with ASD perceive issues that could affect the concept of the parents’ burden and well-being. In recent years, the number of studies carried out on the consequences of caring for people with ASD has notably increased. Sometimes, overload can generate a lack of knowledge about it in families, and the possible resources that they can count on to support them in their day-to-day lives should not be ignored. Ref. [13] affirms that this information will be of vital importance for knowing how to act in daily life situations or what determines if a family finally becomes involved in their children’s adaptation and educational process. For this reason, it is highly important to know not only parents’ opinions, but also their feelings about overload in their work as caregivers. This is to enable visibility for these issues and to promote family members’ well-being and mental health, as these factors also affect people with ASD.

### 1.1. Language in ASD

The definition of ASD is a “specific disorder of the neurobiological and psychological development processes by which the relationship and symbolization functions are constituted” [14] (p. 29). As both functions refer to the basis of human language, it is, therefore, reasonable that both its acquisition and use are affected in people with ASD [7]. According to [8], most of the authors affirm that people with ASD present language impairment. However, this problem depends on a subject’s degree of symptomatology because we can find those who either do not acquire language or acquire it in a very restricted way. Others acquire formal language aspects (phonological and syntactic aspects) but have difficulties with the pragmatic component [15]. Ref. [7] distinguishes between those people who present muteness, whose productions are echolalia from those who learn basic vocabulary and grammar which allows them to understand and produce sentences in a limited way. Finally, some acquire good grammar skills, but have pragmatic and prosodic difficulties. As previously mentioned, Ref. [9] concludes that in 50% of cases, subjects will never develop a functional language. Ref. [16] (p. 59) devise a classification of communication problems that people with ASD present after considering a person’s intellectual and social development: (a) they tend to focus on details (theory of weak central coherence, [17]): they process stimuli by accentuating interest in parts or details and, therefore, perceive information differently from other people. Therefore, as [18] point out, coherence in discourse can be altered in addition to social and pragmatic skills. Ref. [16] indicate that a child understands the meaning of a sentence only from the sum of the parts composing it, and as a result, (s)he is unable to grasp its global meaning; (b) deficit in executive function: responsible for planning, impulse control, inhibition of inappropriate responses, organized search, and flexibility of thought and action. People with ASD seriously alter the planning of behaviors and, therefore, have difficulty with spontaneous language. Ref. [19] attributes this deficit to a maturational alteration of the prefrontal cortex plane of the brain; (c) deficits in the ability to infer, predict and attribute mental states to other people (theory of mind): this involves avoiding social contact, inappropriate relationships from not understanding what other people think, feel or want, and speech disturbances and non-verbal language. According to [20], there may be problems in both expression and understanding language, with difficulties in following orders or responding to demands. Ref. [21] assures from a linguistic point of view that this deficit is interesting because the human ability to transmit information depends largely on the ability to infer it; that is, on recognizing the issuer’s intentions […] Therefore, if someone has such difficulties, the capacity for inference will diminish and, therefore, so will communication; (d) alterations in joint attention: they have problems learning social skills in interaction contexts before starting to speak (laughing, looking into eyes, making sounds, etc.) and they do not build fundamental communicative schemes in language. In relation to this statement, Ref. [19] indicates that it could be due to an alteration in metarepresentational capacity which would manifest itself in difficulties in preverbal communication (e.g., joint attention and proto-declarative acts). To define the linguistic level of people with ASD, a distinction has been made between different levels of affectation.

Thus, research indicates that people with ASD have an alteration in the pragmatic rules that dominate the construction of any conversation, and a literal understanding of the message, no intentionality or interest in communicating, nor any understanding of the sender’s intention [22]. They also have a common pathology in language ability [23,24], lack reciprocity in social discourse, find difficulty in changing the topic of conversation, and use irrelevant details of discourse [3]. They have difficulties with interpreting object pronouns (e.g., him), which is delayed compared to that of reflexives (e.g., himself) [25]. Brief interventions are reduced to “question-answer”; they do not share past personal experiences [21]; they show a deficit in understanding figurative language (humorous elements, prosody, facial expression, gaze, non-verbal language) [26]; and they are unable to capture the illocutionary force of statements. Therefore, they are unable to understand indirect acts, metaphors, jokes and ironies because they have difficulty with processing information [16]. Even children with high-functioning autism (HFA) have pragmatic difficulties compared to children with typical development (TD) [HFA (n = 15) and TD (n = 15) aged 5–9] [27].

Moreover, pragmatic language deficits may lead to a higher risk of anxiety and externalizing disorders due to their pragmatic language deficits [28] (n = 159 ASD children aged 4–7 years) and behavior problems, and hyperactivity and emotional problems [3]. Although they develop verbal language, they still have difficulties with attributing meaning and obtaining effective and functional communication. Therefore, it is necessary to anticipate tasks and transitions through visual and clear supports of activities and the environment to reduce their anxiety and frustration [15].

### 1.2. The Caregiver’s Role: Coping with the Situation

As in people with neurotypical development, people with ASD have different needs and challenges while they develop, but these needs for people with ASD are marked [2]. According to [29], the first big step for parents and caregivers is to accept that a child has difficulties and make the decision to go to a specialist. This diagnostic process is an extensive period that can last months or and even years, and contradictory opinions from different specialists may be received [4]. To face the situations that arise, each family develops a series of coping strategies. Ref. [30], define coping as a “dynamic process that is defined as the set of resources that a subject uses to solve or improve problematic situations, and reduce the tensions that these situations generate. These resources can be beliefs, motivations, social skills, social support and material resources” (p. 577). Various factors, such as socio-cultural context, socio-economic level, level of education, accessibility to different services, etc., strongly influence the implementation of family coping [2]. The study by [31] stated that the family, and specifically parents, are the first nucleus of children’s co-existence and are an important source for detecting problems in children’s development. The family, as [32] assures, becomes the main and most permanent support for individuals. However, Ref. [31] clarify that, although the diagnosis affects the subject’s whole environment, the most affected person is (s)he who acquires the primary caregiver’s role because acquiring a heavier workload drastically modifies the caregiver’s lifestyle and can, consequently, lead to health problems in the short or long term. According to the results of the study by [33], when parents know the diagnosis, they tend to go through a mourning stage when they may note contradictory feelings, and show confusion, sadness, guilt, denial, helplessness and worry. Ref. [34] add that after parents know the diagnosis of someone with ASD, a long and painful process begins which revolves around doubts, fears and anguish in relation to a child’s characteristics, co-existence and future. Hence, the demands related to a child’s disorder are added to the demands that raising a child with TD already entails. For instance, fear of their child being stigmatized, i.e., discrimination and rejection in different contexts [35]. Ref. [36] state that caring for children with different developmental disorders can be demanding for families and parents often have very early concerns about their children’s development which could be aggravated for someone with ASD. According to the Convention on the Rights of the Child in 2009, meeting each child’s individual needs is a primary requirement. In line with this, Ref. [37] state that after the diagnosis a crisis occurs in the family that can be temporary or can prolong over time. This is why social and family support are received at that time. It is true that, as indicated by [32], the treatment approach has been modified because while a subject-centered education by specialists was previously sought, joint collaboration is currently pursued by professionals and families. As ASD is one of the most complex disorders, given its wide variability of symptoms and needs, each family must adopt education guidelines based on their child’s characteristics. These will influence the well-being of both the child and his/her family [2]. For this purpose, a series of professional support services must be properly informed and guaranteed to ensure the quality of life and psychological well-being of both people with ASD and their family environment.

#### 1.2.1. The Importance of the Caregiver’s Role

Due to the symptoms of people with ASD, the presence of a caregiver is necessary to a greater or lesser extent depending on the severity of each case. Ref. [38] differentiate two caregiver types: formal caregivers, that is, healthcare personnel outside the family nucleus; and informal caregivers. These are generally family members who play the role of the subject’s caregiver. We should bear in mind that in this second case, apart from the contribution made by the subject’s caregiver, the presence of an additional caregiver might be also sought to preserve the individual’s health. When considering the consequences for a caregiver, we must first distinguish between the objective burden and the perceived burden. As seen in the study by [39], subjective burden refers to the attitudes and emotional reactions generated before the experience of having to take care of a family member, and it is related to the way in which the situation is perceived. The objective burden is related to the performance of the caregiver’s role and the modifications (s)he must make in his/her daily life. Ref. [40] emphasize that the caregiver’s feeling of burden is considered an important factor in the caregivers’ quality of life and for his/her well-being. The World Health Organization in 1996 defines quality of life as “the perception that an individual has of his place in existence, in the context of the culture and value system in which he lives and in relation to their objectives, their expectations, their norms and concerns” (p. 385). Therefore, the quality of life of not only subjects with ASD, but also of their families, is one of the main objectives to be considered for adequate intervention because as [32] points out, people with ASD require more attention given the complexity of their demands and, consequently, family overload might occur.

#### 1.2.2. Support to Families

Once children are diagnosed with ASD, parents often go through a series of grief phases. For this process to be more bearable and to be able to better manage their emotions, parents should seek information and support from appropriate sources and professionals because the effectiveness of the actions they take and their decisions about their children depend on this [13]. As determined in the study by [41], helping caregivers cope with stress and burden, as well as developing good communication skills, can help to protect against cognitive decline and to even improve the cognitive performance of subjects and their family. However, as [36] claim, many parents report their disagreement with the lack of help received from professionals after the diagnostic process. For this reason, Ref. [32] emphasizes the fundamental need to provide support and guidance to families with people with ASD to either help parents in the adaptation process and with emotional acclimatization, or to facilitate parents’ roles in their children’s education, especially in the early years. In another study, Ref. [37] mention that communication with other families in similar situations is an important tool. Seeking alternatives coping tools in a group setting is proposed as a liberating decision because, sometimes, parents must attempt to generate support among themselves if they cannot obtain support from other services. As [42] state, it is important to accurately measure the impact that caring for children with ASD has on their parents, because parental stress, coping strategies and perceived quality of life are independent of the child’s disorder [2].

Regarding the consequences for caregivers, Ref. [43] points out that the main caregiver of someone with ASD can experience relevant changes in their usual tasks and responsibilities. Therefore, different areas of their personal lives may be affected, including: social relationships, economic status due to working few hours or losing a job, or even their psychological capacity, which is why considerable social support is necessary to attempt to maintain family stability [44]. The WHO in 2017 points out that the socio-economic effects of ASD are important because experiencing this disorder can significantly limit a person’s ability to function socially and thus, to generate an impact on subjects’ educational and social achievements, and on their job opportunities. Another aspect to contemplate is the caregiver’s subjective happiness. Although happiness throughout history has had different conceptions, it currently includes the notions of subjective well-being, psychological well-being, and health or personal growth [45]. In other words, it is a paradigm that refers to the most global version of the valuation of life and is, in turn, inclusive with the different perspectives of well-being today [46]. As Ref. [47] indicates, quality of life in relation to someone’s well-being arises. Moreover, not only does it refer to the subjective degree of an individual’s satisfaction, but also to the resources that an individual can access to deal with any problems that arise. This implies that the family is the main nucleus that is responsible for generating a stable quality of life and ensuring the well-being of different members [48]. Furthermore, as Ref. [49] points out, it is true that everyone has a different perspective on well-being. Further, everyone is aware of this. Therefore, as a consequence of ASD symptoms (such as difficulty communicating, mental inflexibility and/or difficulties in the social area), caregivers could suffer worsening physical and mental health [2]. As Ref. [50] postulate, mothers’ mental health, defined as their well-being, would be closely related to the severity of the child’s disability. According to [51], the perceived quality of life of the relatives of children with ASD is influenced by the differences in their children’s level of comprehension, language and communication. Along these lines, Ref. [50] suggest that emotional stress in the parents of children with ASD is negatively associated with children’s communication skills, to which [52] add that verbal children present more communication problems and antisocial behaviors, but nonverbal children tend to isolate themselves and experience difficulties in social relationships. Ultimately, as indicated by [42], other determining factors can be added to the caregiver’s life that sometimes go unnoticed, such as a lack of time to satisfy their own needs, financial difficulties caused by medical care, therapies, or even caregivers losing their job. Therefore, it is essential to study the caregiver’s burden, in order to correctly intervene with the subject and ensure the well-being of both parents and children [53]. Furthermore, as [2] state, we must not forget that quality of life is considered an essential aspect of health and treatment (improving the patient-therapist relationship, assessing the effectiveness of different treatments, evaluating health services, etc.), for which it is be fundamental to treat the subject’s condition. As stated by [44], the quality of life of families with ASD is altered by the usual problems. These problems are in addition to the consequences of the subject’s symptom severity (such as communication problems, behavior problems, self-harm, rituals and intense frustration, etc.). Therefore, undoubtedly, a series of consequences will appear in caregivers (such as stress, depression, and misunderstanding). Studying these consequences in caregivers is a relevant tool in preserving their health and maintaining their quality of life, because the family’s active participation in the development process of the child with ASD does not imply ignoring the study of the family’s support needs [32].

After justifying and objectifying the study, we asked the following research question: does the level of severity and the pragmatic aspects of language influence the overload and well-being of caregivers of children with ASD? To answer this question, we consider the studies carried out by [51] which result in a close link between the different variables. In this way, we consider the following hypotheses:
**Hypothesis** **1.***The heavier the perceived burden, the worse the caregiver’s well-being.*
**Hypothesis** **2.***The more severe the ASD case and the fewer the pragmatic aspects, the worse the caregiver’s well-being.*
**Hypothesis** **3.***The more severe the level of the person with ASD and the fewer the pragmatic aspects, the heavier the perceived burden.*
**Hypothesis** **4.***The influence of ASD severity of the person with ASD and the level of pragmatic aspects in overload are moderated by well-being.*

## 2. Materials and Methods

### 2.1. Participants

The selected sample consisted of 28 user families of the Huesca Autism Association, and families from the province with children with ASD. Of these, 10 participants were men which represented 35.7% of the sample, and 18 were women (64.3%). The participants’ ages ranged from 32 to 67 years, with a mean age of the study participants being 44.43 years (SD = 11.65).

### 2.2. Measures

The instruments used to measure the variables included in this research work are detailed below:

*Severity level*. This was obtained from the Quantitative Checklist for Autism in Toddlers (Q-CHAT) [54]. It consists of 25 items that address parents and allows them to quantify their children’s autistic traits at an early age (from 18 months and can be administered up to 40 months). It takes between 5 and 10 min to complete the questionnaire. It is answered according to a 5-point Likert scale (0–4). The preliminary data on the psychometric properties of the original Q-CHAT version [54] show acceptable internal consistency (Cronbach’s alpha = 0.83) and a test-retest correlation of 0.82 (n = 330). The questionnaire has been translated into Spanish and adapted in Colombia [4,55]. This is the version that was used in the present study.

*Caregiver overload*. This was obtained using the [56] Caregiver Burden Scale. This instrument quantifies the degree of overload experienced by caregivers of dependent people. It can also be applied to the relatives of children with autism.

*Well-being scale*. This was measured from the Subjective Happiness Scale of [57]. It is a global measure of subjective happiness that consists of four items with a Likert-type response to evaluate a well-being category as a global psychological phenomenon by considering the definition of happiness from the respondent’s perspective.

*Pragmatic language*. This was taken from the adaptation into Spanish, carried out by [58], of the original Children’s Communication Checklist (CCC) by [59]. The original questionnaire consists of 70 questions, but the first 15 questions were eliminated because they assess aspects related to the form of language (phonology and morphosyntax, which are usually evaluated with specific tests) and not to its functional use. In this way, the Spanish adaptation is a questionnaire with 55 questions related to the social (pragmatic) use of language and paralinguistic social communication. It is completed by parents with these answers: ‘no’, ‘sometimes’ or ‘yes’.

### 2.3. Data Collection Procedure

To carry out this study, first, the central theme of this work was defined. Next, the questionnaires that were in accordance with the theme were selected and reflection was made on which participants could be given the tests. Later, both the Association and the families were contacted to report the desired work by explaining in detail the objective and the procedure to be followed at all times.

Participation was completely voluntary and the data remained confidential and anonymous (socio-demographic information was requested: age, sex, etc.), although no data were requested that would allow the subject answering the questionnaires to be identified. The battery of instruments was independently completed to be later delivered by phone.

Data were collected in an Excel table. They indicated the number of members of the participating families, the participant’s sex (with 0 being male and 1 being female), and the answers obtained in the four questionnaires.

### 2.4. Data Analysis

In this work, several data analyses were carried out to test the hypotheses. First, interactions were measured, such as the relation between the dependent and independent variable and if it could be modulated by a moderating dependent variable. For this purpose, the Hayes method was used [60] which is described on this website: http://afhayes.com/introduction-to-mediation-moderation-and-conditional-process-analysis.html, (accessed on 10 January 2022). The lower significance of *p* < 0.10 indicated the marginal significance of the studied interaction. In the interaction effects, the conventional limit at the *p*-level was 0.10. This *p*-level has been suggested by several researchers, such as [61], to protect the test from the probability of making a Type II error when testing and performing modulation analyses. In this way, the results presented in the following section were obtained by means of these statistical programs.

## 3. Results

The descriptive analyses (means and standard deviations) of each studied variable, and the correlations between them are calculated (see Table 1).

Hypothesis 1 proposed that the heavier the perceived burden, the worse the caregiver’s well-being. The results showed a negative relation between overload and well-being (r = −0.46, *p* < 0.05). Therefore, H1 was accepted.

Hypothesis 2 proposed that the more severe the ASD case and the fewer the pragmatic aspects, the worse the caregiver’s well-being. The results showed that pragmatics and well-being did not correlate (r = 0.10, ns). However, the relation between severity and well-being was negative (r = −0.39, *p* < 0.05). Therefore, H2 was partially accepted.

Hypothesis 3 proposed that the more severe the symptoms of the person with ASD and the fewer the pragmatic aspects, the heavier the perceived burden. The results revealed that pragmatics and overload did not correlate (r = −0.02, ns). Likewise, the relation between severity and overload was negative (r = 0,16, ns). Therefore, H3 was rejected.

Hypothesis 4 proposed that the influence of the ASD severity of the person with ASD and the level of pragmatic aspects in overload are moderated by well-being. The results indicated that the relation between the severity of the person with ASD and the caregiver burden was negative when well-being levels were high, neutral when well-being levels were medium and positive when caregiver well-being levels were low (see Table 2). In other words, the worse the severity, the greater the perception of overload if the caregiver had low well-being levels (B = −0.89, *p* < 0.05). This relation between severity and overload modulated by well-being occurs regardless of the level of pragmatics (see Figure 1) which is why H4 was partially accepted.

## 4. Discussion

### 4.1. Contributions

The aim of this study was to analyze the moderating role of parental well-being and the ASD pragmatic level in the relation between ASD severity and parental overload. Hypothesis 1 proposed that the heavier the perceived burden, the worse the parent’s well-being. The results showed a negative relation between overload and well-being. Therefore, H1 was accepted. In the study by [32], the family’s role in the quality of life and self-determination of people with ASD ensures that these subjects need much more attention because of the complexity of their demands and, therefore, as support needs increase, a case of family overload might occur. Along the same lines, Ref. [38] wrote the article, “Cardiac response to cognitive stress in informal caregivers of people with Autism Spectrum Disorder” with a sample of 72 participants. They concluded as a result that fulfilling the caregiver role is a source of stress that can have significant physical and mental health consequences over time. This has a negative impact on parents’ quality of life because new social, physical and psychological demands constantly appear. In contrast, authors such as [62] conclude that family members can be an important source of support, but rejection may also arise, directly affecting the subject and their development. However, authors such as [63] conclude that a considerable number of families who have children with autism show resilience factors, and report that they have been strengthened as a result of a member of the family having a disability.

Hypothesis 2 proposed that the more severe the ASD case and the fewer the pragmatic aspects, the worse the caregiver’s well-being. The results showed that aspects of language pragmatics and well-being did not correlate. However, the relation between severity and well-being was negative. Therefore, H2 was partially accepted. This result falls in line with the study by [38] who asserted that a more severe autistic symptomatology increases the proliferation of stress, and creates a consequent increase in the anxious state (depressive mood). This, therefore, leads to worse caregiver health and quality of life. Ref. [51] conducted a study entitled “Language comprehension disorders in non-verbal children with autism spectrum disorders and their implications for family quality of life”. It concluded that the perceived quality of life in relatives of children with ASD and TD can be explained partly by differences in their children’s level of comprehensive language and communication. Ref. [64] point out in their study with 301 primary caregivers of children with autism that there are several reasons why a caregiver’s mood can be associated with ASD severity: the quality of uneducated counseling services, the demand for better qualified and trained staff, and the need for new treatment approaches. Ref. [50] add that maternal health is related to the extent of a child’s general behavioral problems and not to the degree of autistic symptoms. On the contrary, Ref. [65] in their study entitled “Experiences of relatives of critical care. Nursing in critical care” conclude that family members’ stress levels can also affect the patients’ condition and their development. Therefore, lowering and treating family stress levels should be a priority because it could improve the subject’s recovery process.

Hypothesis 3 proposed that the more severe the symptoms of the person with ASD and the fewer the pragmatic aspects, the heavier the perceived burden. The results showed that pragmatic aspects of language and perceived overload did not correlate. Likewise, the relation between severity and perceived overload was negative. Therefore, H3 was rejected. Despite the obtained results, various studies emphasize that subjects’ development level in different areas determines the overload perceived by caregivers. Indeed, Ref. [2] in their study, “Coping, parental stress and quality of life of the main caregivers of people with ASD” which was carried out with a sample of 50 caregivers of people with ASD, offers the result that difficulty communicating, mental inflexibility and/or difficulties in social areas determine the burden perceived by caregivers and, therefore, their physical and mental health. Similarly, Ref. [66] assure that serious deficits in communication and social relationships, frequent disruptive behaviors, self-injurious behaviors, and repetitive and stereotyped behaviors, are factors that definitely alter the family and social life of a caregiver. On the contrary, authors such as [67] conclude that accepting a dependent person’s communication difficulties is essential to manage meeting the needs of both the individual and his/her family.

Hypothesis 4 proposed that the relation between the symptom severity of people with ASD and its influence on their parents’ overload is moderated by parental well-being and ASD pragmatic level. The results revealed that the relation between severity and parental overload was positive when parental well-being levels were low, neutral when well-being levels were medium, and negative when parental well-being levels were high. Therefore, the overload of the parents of people with autism depends on parents’ well-being and not on the severity of the symptoms of people with ASD. That is: the more severe the symptoms of people with ASD, the greater the perception of overload, but only if the caregiver has low well-being levels. Moreover, this relation of gravity and overload moderated by well-being occurs regardless of the level of pragmatics. This is why H4 was partially accepted. Despite the obtained results, the study by [51] concludes that improving comprehensive language skills or limitations in nonverbal communication in children with ASD could alleviate or substantially reduce the perceptions of dissatisfaction with quality of life in their families. In contrast, other studies conclude that depressive symptoms and high anxiety levels are two characteristic patterns of parents of people with ASD, most of which are linked to various contextual and psychosocial variables which can modulate this symptomatology. To conclude, Ref. [63] assures in his study that several parents stressed that their parenting experiences were not so different from raising any other child without disabilities.

### 4.2. Limitations and Study Proposals

This study has some limitations that should be considered in future works. First, it is important to highlight our sample size (28 relatives with ASD). This could have limited the possibility of acquiring significant data to confirm or reject the hypotheses. In other words, the study could be affected by a type II error. It is important to point out the difficulty of carrying out a study with selected participants due to the wide variety of ASD symptoms. In addition, the study is not a longitudinal one and therefore, no causal relations can be established because it is a correlational study. Another limitation is the common variance of the method due to the use of questionnaires. This should also be considered in future studies to correct any biases that affect the relations between constructs.

### 4.3. Future Research

Next, several future research lines are proposed based on this work. It is worth mentioning that, although data are informative, our sample size does not allow inferences to be made to the general population. Thus, future research should work with a bigger sample size and include children from different population groups and socio-economic strata. Future studies should also bear in mind other aspects, such as: how the mother and father view the situation, the variation in resources and the support received at the time of the child’s birth and, if possible, future work should also obtain data from the person with ASD about their satisfaction, the results obtained in the intervention, etc.

## 5. Conclusions

Despite the results obtained showing that the responsibility for parental overload depends more on their well-being than on the severity of the person with ASD, it was confirmed that being the main caregiver of a person with ASD generates a high stress level with effects to a greater or lesser extent on all areas of the person’s life (social, emotional, occupational, etc.). Therefore, health consequences may arise that affect their own quality of life in both the short and long term which can lead to family overload situations. For this reason, throughout this work, we have attempted to explain that the way in which the situation of having a child with ASD is faced depends on different factors (individual, family and environment). However, it is important to highlight that the caregiver’s stress level is mainly due to the coping styles and strategies that they acquire. It is also based on their knowledge and on the services and support that they receive from the relevant community to cope with the crisis that may arise when caring for a child with ASD. Ruiz et al. (2017) assure that social support is essential when facing any case of disability because it cushions the impact of chronic stress on health by helping to reduce negative consequences on quality of life.

Having a child with a disability has an impact on any family. For this reason, with this study we aim to make visible these families’ experience of constant struggle and the obstacles they face every day to achieve the highest possible development level for their children. Despite this disorder only recently becoming relevant in social terms, lack of true information about its symptoms continues to be one of the main reasons for the stigmatization and discrimination of ASDs. It is essential for families to feel supported and sheltered in order to face this diagnosis and the demands that their children require throughout their vital development. Not only is the intervention of all professionals essential, but the commitment of families is too, as well as the participation of society as a whole. Only in this way can the full inclusion of people with ASD be achieved in society because unlike what many people think, “Having a child with a disability does not necessarily imply a sentence for the family. There are many families that advance constructively and are strengthened despite the adverse circumstances of life” [68].

## Figures and Tables

**Figure 1 children-09-00769-f001:**
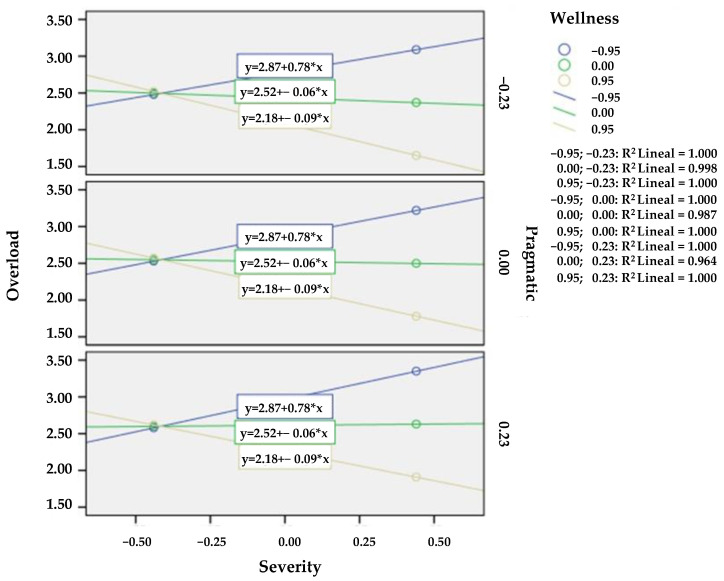
Results of the regression analysis for overload.

**Table 1 children-09-00769-t001:** Descriptive statistics and correlation analysis.

	*M*	*SD*	1	2	3	4	5	6
1. Pragmatic	0.92	0.23	1					
2. Overload	2.66	0.67	−0.02	1				
3. Wellness	3.76	0.95	0.10	−0.46 *	1			
4. Severity	2.94	0.44	−0.03	0.16	−0.39 *	1		
5. Age	44.43	11.65	0.33	0.04	−0.08	−0.20	1	
6. Sex	0.64	0.49	0.17	−0.07	0.11	0.05	−0.08	1

Note. * Correlation is significant at the −0.05 level (2-tail).

**Table 2 children-09-00769-t002:** Results of the regression analysis for overload (Over).

Variable	B	R^2^	ΔR^2^
Sex	0.3		
Age	0.01		
Pragmatic (Prag)	0.38		
Severity (Sev)	−0.05		
Wellness (Well)	−0.37 *		
Sev × Well	−0.89 *		
Sev × Prag	0.37	0.39 ^†^	0.18 ^†^

Note: B are the non-standardized regression coefficients ^†^ < 0.10, * *p* < 0.05.

## Data Availability

The data presented in this study are available on reasonable request from the corresponding author. The data are not publicly available due to privacy restrictions.

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
