# Peer review of "The Influence of ASD Severity on Parental Overload: The Moderating Role of Parental Well-Being and the ASD Pragmatic Level"

_children, 2022, doi:10.3390/children9060769_

Round 1

Reviewer 1 Report

We suggest a more formal approach to the issue; thus, we invite to review some sentences or to summarize some paragraphs

(i.e. As Jolliffe, Landsdown and Robinson (1992, p. 15) stated, “I am not looking at the whole but rather just the outline or the part. I cannot look at a picture completely, but only a small section at a time”)

(i.e.A brief history of autism)

References are needed:

World Health Organization (WHO, 2021), it is estimated that 1 in 160 children has ASD and in recent years various studies have proven that there is a gradual increase in cases .

Was the informed consent obtained from all parents?

Could you provide results of the administration of well-being questionnaires to a group of parents with neurotypical children or children with neuropsychiatric disorder other than ASD?

How opening questions are integrated in this study in order to convey standardized results?

Have children with autism been evaluated by experienced neuropsychiatrists in order to assess degree of functioning/severity or by standardized diagnostic tools ? Bias could occur from questionnaires parent mediated regarding this aspect.

How do you know that wellness affects load of the situation and not vice versa? The conclusion is quite vague about it.

Author Response

Specific Responses to Reviewers’ Comments on

The influence of ASD severity on parental overload: The moderating role of parental well-being and ASD pragmatic level

Manuscript ID children-1623277

The reviewer indicated that the manuscript is interesting and relevant. Moreover, the study highlights the relevance of parenting wellbeing when providing services to children with ASD and their families, presenting a clear overview of a small but useful building block of empirical evidence concerning ASD. In this sense, our manuscript could potentially contribute to ASD and their families’ research. The reviewer also offered a number of practical ways in which he/she believes that it can be improved, although he/she indicated that none of them is intended to suggest that the research is seriously flawed. We have used these comments and suggestions in a straightforward way to improve the manuscript. Thank you in advance for these previous words and all comments, suggestions and change proposals.

Reviewer(s)' Comments to Author:

Reviewer: 1 Comments:

We suggest a more formal approach to the issue; thus, we invite to review some sentences or to summarize some paragraphs

(i.e. As Jolliffe, Landsdown and Robinson (1992, p. 15) stated, “I am not looking at the whole but rather just the outline or the part. I cannot look at a picture completely, but only a small section at a time”) (i.e.A brief history of autism)

Following Reviewer#1’s comments, we have eliminated the text and reference.

Jolliffe, T., Landsdown, R., & Robinson, C. (1992). Autism: A personal account. Communication, 26, 12-19 (See pages 1 and 17). Moreover, we have eliminate the paragraph titled “A brief history of autism”

References are needed:

World Health Organization (WHO, 2021), it is estimated that 1 in 160 children has ASD

and in recent years various studies have proven that there is a gradual increase in cases .

The World Health Organization (WHO) on June 1, 2021 publishes on the following website; https://www.who.int/news-room/fact-sheets/detail/autism-spectrum-disorders

"Autism spectrum disorders". However, the source of the data is from the following reference. :

Elsabbagh, M., Divan, M., Koh, Y. J., Kim, Y. S., Kauchali, S., Marcín, C., Montiel-Nava, C., Patel, V., Paula, C., Wang, C., Yasamy, M., and Fombonne, E. (2012). Global Prevalence of Autism and Other Pervasive Developmental Disorders, Autism Research, 5(3), 160–179.

World Health Organization. (1 june 2021). Autism spectrum disorders. https://www.who.int/news-room/fact-sheets/detail/autism-spectrum-disorders.

We agree with reviewer #1 to include both references:

Was the informed consent obtained from all parents?

Could you provide results of the administration of well-being questionnaires to a group of parents with neurotypical children or children with neuropsychiatric disorder other than ASD?

In most studies, comparisons are made between parents of neurotypical children or in comparison with other disorders. In this case, the study focuses only on ASD.

How opening questions are integrated in this study in order to convey standardized results?

Now we have eliminate in the section of "method" Open questions related to aspects such as sociodemographic information, the detection process or the services received. In fact, we are not analyze this information in this study. De hecho, no estamos analizando esta información en este estudio.

Have children with autism been evaluated by experienced neuropsychiatrists in order to assess degree of functioning/severity or by standardized diagnostic tools? Bias could occur from questionnaires parent mediated regarding this aspect.

The interest of the study is not so much to know the diagnosed severity, but the severity perceived by the parents. In fact, it could be that a person has a more severe diagnosis and the parents perceive less severity of symptoms. In any case, people belong to an Association and are knowledgeable about the specific case; it is possible that their perception of severity is based on the actual diagnosis.

How do you know that wellness affects load of the situation and not vice versa? The conclusion is quite vague about it.

The correlation between wellness and load is negative (r=-,46, p<.05), for this reason, the aim of this study is to analyze other variables that could modulate this negative relationship.

References cited in the answers to the comments from the reviewer

Elsabbagh, M., Divan, M., Koh, Y. J., Kim, Y. S., Kauchali, S., Marcín, C., Montiel-Nava, C., Patel, V., Paula, C., Wang, C., Yasamy, M., and Fombonne, E. (2012). Global Prevalence of Autism and Other Pervasive Developmental Disorders, Autism Research, 5(3), 160–179.

World Health Organization. (1 june 2021). Autism spectrum disorders. https://www.who.int/news-room/fact-sheets/detail/autism-spectrum-disorders.

Hayes, A. F. (2017). Introduction to mediation, moderation, and conditional process analysis: A regression-based approach. Guilford publications.

Reviewer 2 Report

This study aims to reveal the relationship between the severity of the ASD and the overload of their parents, by analyzing the moderation effect of parental well-being and ASD pragmatic level. I appreciate that this study highlights the relevance of parenting wellbeing when providing services to children with ASD and their families. Though the topic is interesting and relevant, there are major concerns in methodology and therefore were unable to draw conclusive results. There are also issues with writing and reporting the data. The main points are listed below.

Introduction: The authors provided much information on the background and extensive reviews of literature, such as the history of ASD and different aspects of language in ASD, which made it hard to link to the study aims directly.

Participants: the sample only include 28 families (in the abstract, the sample size was 29, might be an error?), which draws concern of low power. The authors did not provide any power analysis to justify their sample size selection and did not describe what was the sample’s characteristics based on the results of the measures. The recruiting process is not clear, did you intend to recruit children with a variety of severity since this is one of the important independent variables?

Analysis: The authors seemed to perform a series of correlation analyses, regression analysis, and moderation analysis by Hayes (2013). This was not clearly presented in the text.

Results: Table and Figure are not indicated in the text. The relationships between pragmatics and severity (independent variables) and overload (dependent variables) were not significant, how could you place them in the regression analysis? Even you did, these variables did not predict overload, and it’s not meaningful to further examine the moderating effect of wellbeing. The non-significant results may be due to the small sample size and low power, the authors are encouraged to increase numbers and improve their data presentation. 

Author Response

Specific Responses to Reviewers’ Comments on

The influence of ASD severity on parental overload: The moderating role of parental well-being and ASD pragmatic level

Manuscript ID children-1623277

The reviewer indicated that the manuscript is interesting and relevant. Moreover, the study highlights the relevance of parenting wellbeing when providing services to children with ASD and their families, presenting a clear overview of a small but useful building block of empirical evidence concerning ASD. In this sense, our manuscript could potentially contribute to ASD and their families’ research. The reviewer also offered a number of practical ways in which he/she believes that it can be improved, although he/she indicated that none of them is intended to suggest that the research is seriously flawed. We have used these comments and suggestions in a straightforward way to improve the manuscript. Thank you in advance for these previous words and all comments, suggestions and change proposals.

Reviewer(s)' Comments to Author:

Reviewer: 2 Comments:

Introduction: The authors provided much information on the background and extensive

reviews of literature, such as the history of ASD and different aspects of language in ASD,

which made it hard to link to the study aims directly.

Following Reviewer#2’s comments, we have eliminate the paragraph titled “A brief history of autism”

Participants: the sample only include 28 families (in the abstract, the sample size was 29,

might be an error?), which draws concern of low power. The authors did not provide any

power analysis to justify their sample size selection and did not describe what was the

sample’s characteristics based on the results of the measures. The recruiting process is not

clear, did you intend to recruit children with a variety of severity since this is one of the

important independent variables?

The selected sample consisted of 28 user families of the Huesca Autism Association, and by families from the province with children with ASD. We have modified this data in the abstract section. The manuscript focuses on a growing ASD association. Moreover, the increase in users of the association in the last three years has gone from 16 families to 74. Nevertheless. Thus, the participation in the study focuses not so much on new families joining the association as on families who are in a different process from families with long membership in the organization. Therefore, the participation of one third of the association seems to us to yield relevant information. The study does not focus on the analysis of ASD severity but on the perception of ASD severity by parents of people with ASD.

Analysis: The authors seemed to perform a series of correlation analyses, regression

analysis, and moderation analysis by Hayes (2013). This was not clearly presented in the

text.

The manuscript indicates the Hayes method web page (new link) where the method is fully explained, as well as the free download of the necessary software in SPSS to be able to analyze the results. http://afhayes.com/introduction-to-mediation-moderation-and-conditional-process-analysis.html

Furthermore, we have also included a new reference:

Hayes, A. F. (2017). Introduction to mediation, moderation, and conditional process analysis: A regression-based approach. Guilford publications.

Results: Table and Figure are not indicated in the text. The relationships between

pragmatics and severity (independent variables) and overload (dependent variables) were

not significant, how could you place them in the regression analysis? Even you did, these

variables did not predict overload, and it’s not meaningful to further examine the

moderating effect of wellbeing. The non-significant results may be due to the small sample

size and low power, the authors are encouraged to increase numbers and improve their data presentation.

As stated the reviewer#2 and the MS indicated (MS# p. 9), the lower significance of p<.10 shows us a marginal significance of the studied interaction. In interaction effects, the conventional limit on the p-level is .10. This p-level has been suggested by several researchers, such as Champoux and Peters (1987), to protect the test from the probability of making a Type II error when testing. perform modulation analyses. Nevertheless, in this case, the interaction effect between severity and wellbeing over overload is (B=-,89, p<.05)

We have indicated the tables and figures in the text of the manuscript.

In closing, we very much appreciate these thoughtful suggestions. As always, let us know if more work is needed to further improve our manuscript.

References cited in the answers to the comments from the reviewer

Elsabbagh, M., Divan, M., Koh, Y. J., Kim, Y. S., Kauchali, S., Marcín, C., Montiel-Nava, C., Patel, V., Paula, C., Wang, C., Yasamy, M., and Fombonne, E. (2012). Global Prevalence of Autism and Other Pervasive Developmental Disorders, Autism Research, 5(3), 160–179.

World Health Organization. (1 june 2021). Autism spectrum disorders. https://www.who.int/news-room/fact-sheets/detail/autism-spectrum-disorders.

Hayes, A. F. (2017). Introduction to mediation, moderation, and conditional process analysis: A regression-based approach. Guilford publications.